# Drivers and dynamic mechanisms of sports tourism integration in cross-border regions: Evidence from the Guangdong-Hong Kong-Macao Greater Bay Area

Jun Yuan👤*, Canyu Chen, Zhouxin Luo, Yanhong Liu*

School of Physical Education, Shenzhen University, Shenzhen, China

* yuanjun@szu.edu.cn (JY); 841226273@qq.com (YL)

## Abstract

The integration of sports tourism industry plays a crucial role in fostering high-quality regional economic growth, enhancing industrial coordination, improving quality of life and promoting sustainable tourism development, particularly in cross-border regions. However, existing research has predominantly focused on macro-level policy and market analyses, paying limited attention to micro-level drivers and underlying mechanisms. To address this gap, this study adopts a mixed-methods approach, using the Guangdong-Hong Kong-Macao Greater Bay Area (GBA) as a case study. Grounded Theory is first employed to identify key drivers and construct a conceptual model of their dynamic mechanisms. Subsequently, Structural Equation Modeling (SEM) is applied to empirically test the model. The results confirm six significant drivers that positively influencing integration outcomes: government behavior, resource environment, enterprise development, market demand, technological innovation, and human capital. Notably, the analysis reveals substantial differences in their impact strength. Human capital ($\beta = 0.619$, $p < 0.001$) and technological innovation ($\beta = 0.541$, $p < 0.001$) emerge as the dominant drivers, followed by enterprise development needs ($\beta = 0.321$, $p < 0.001$). In contrast, market demand ($\beta = 0.065$, $p < 0.001$), resource environment ($\beta = 0.041$, $p = 0.017$), and government behavior ($\beta = 0.052$, $p = 0.002$) show the weakest direct effects, with government behavior further constrained by policy fragmentation. To facilitate high-quality and sustainable integration, targeted policy recommendations are proposed, including enhancing cross-border policy coordination and institutional alignment, leveraging technological innovation through digital integration platforms, strengthening enterprise-led collaboration and cross-border industrial clusters, cultivating and mobilizing interdisciplinary talent across the region, and expanding segmented markets through differentiated regional branding. This study contributes to the theoretical framework of sports tourism integration by empirically elucidating the interplay and relative strength of the drivers and

**Data availability statement:** All original data files are available from the figshare database (DOI: 10.6084/m9.figshare.28377632).

**Funding:** This work was supported by the National Social Science Foundation of China (20BTY054).

**Competing interests:** The authors have declared that no competing interests exist.

their dynamic mechanisms in cross-border contexts. Furthermore, it offers actionable insights for policymakers and industry stakeholders in the GBA.

## Introduction

With the rapid growth of social consumption and the increasing demand for sports tourism, the sports tourism industry has become an important driving force for promoting high-quality regional economic development [1]. The sports tourism industry integrates the two emerging sectors of sports and tourism, not only having a direct effect on economic growth but also contributing to the enhancement of regional cultural soft power, optimization of resource allocation, and promotion of green development, thereby boosting the overall regional competitiveness from multiple dimensions [2]. The Guangdong-Hong Kong-Macao Greater Bay Area (GBA), as an critical strategic area for China's economic and social development, represents a typical Cross-Border Region. It is composed of three geographical and administrative divisions: the nine Pearl River Delta municipalities in Guangdong Province, the Hong Kong Special Administrative Region and the Macao Special Administrative Region (Fig 1). This region exhibits unique cross-border characteristics in culture, policy and economy, forming a complex and organic regional cooperation system, which plays a vital role in the national economic layout and regional coordinated development.

Currently, the sports tourism industry in GBA has formed an initial development model centered around large-scale sports events, combining regional cultural characteristics. For example, events such as the Guangzhou Marathon and the Shenzhen China Cup Sailing Regatta have attracted a large number of domestic and international visitors, driving the dual growth of sports and tourism consumption in the region. Meanwhile, a number of region-specific sports tourism projects have gradually emerged, such as the Guangdong-Hong Kong-Macao Greater Bay Area Outdoor Leisure Sports Festival and the Guangdong-Hong Kong-Macao Greater Bay Area Dragon Boat Race, showcasing the potential for the integration of sports and tourism resources in the region. However, despite the GBA's significant advantages in policy support, resource endowment, and market potential, the sports tourism industry still faces challenges such as product homogeneity, lack of innovation, and inadequate resource-sharing mechanisms [3]. Addressing these issues is key to achieving high-quality integrated development. Notably, the Greater Bay Area is about to jointly host the 15th National Games, which will undoubtedly bring new development opportunities to the sports tourism industry in the region [4]. As the highest level comprehensive sports event in China, the National Games will not only attract a large number of athletes, coaches and sports fans to participate, but also greatly enhance the international visibility and influence of the GBA, further promote the deep integration of sports and tourism, and promote the sustainable development of regional sports tourism.

Many scholars have extensively and deeply discussed the integration and development of the sports tourism industry, analyzing the driving forces, models, mechanisms, and paths of integration from various perspectives, and offering numerous

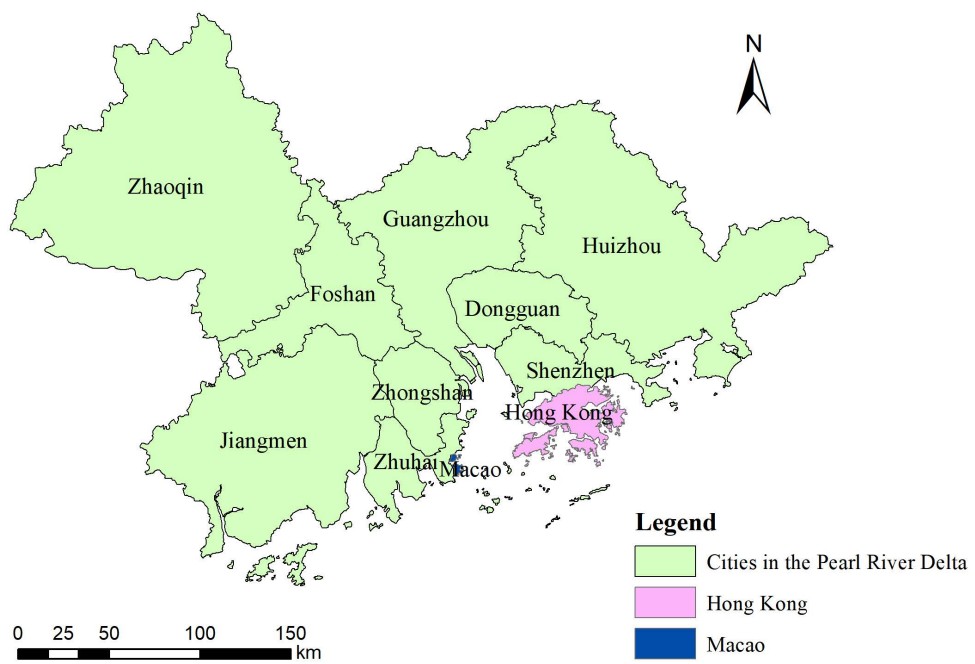

**Fig 1. Major cities in the Guangdong-Hong Kong-Macao Greater Bay Area.**

valuable insights and recommendations [5–14]. Regarding driving factors, scholars have conducted broad research on the multidimensional factors driving the integration of the sports tourism industry [15–19]. For instance, Hinch and Higham identified policy, market, and technology as the three core factors driving the integration of sports tourism, with policy support playing a particularly effective role in facilitating the integration and utilization of industry resources [6]. Ratten analyzed sports tourism projects in different regions and found human capital and technological innovation as key enablers of product diversification [10]. Additionally, Wang et al. conducted an empirical analysis of the sports tourism industry in the southeastern coastal region of China, identifying government policy support, the diversity of market demand, and enterprise innovation capabilities as the main drivers of the region's sports tourism development [17]. In terms of technological innovation, Mao and Li utilized big data analysis to reveal the role of technology in advancing the sports tourism industry [11]. Their research showed that the application of intelligent management systems and virtual reality technologies not only enhances user experiences but also fosters inter-industry synergies. Meanwhile, market demand has exhibited a trend of diversification, particularly in sub-sectors such as health tourism and experiential tourism, where consumer demand is constantly evolving. This, in turn, drives innovation in enterprises and the integration of sports tourism resources. A review of the above literature reveals that, while substantial research has been conducted on the macro-level factors influencing the integration of sports tourism, there remains a significant gap in research on micro-level factors.

In terms of research methods, existing studies on the factors influencing sports tourism integration primarily adopt qualitative research methods [16,17], while a few quantitative studies rely on statistical yearbook data and employ methods such as regression analysis [18,19], gray relational analysis [20], and factor detection via geographic detectors [21] to identify the macro-level factors driving sports tourism integration. However, there is a lack of quantitative studies based on enterprise interviews and survey questionnaires, which construct econometric models. Direct enterprise surveys and data analysis enable a more precise capture of market dynamics and enterprise needs, providing a more robust empirical foundation for theoretical research. Overall, the current academic research on the driving factors of sports tourism industry

integration remains in an exploratory stage, with a narrow focus, limited methodological diversity, and a significant lack of empirical research at the micro level.

Building on the above background, this study addresses the following key questions: (1) What are the critical driving factors shaping the integration of the sports tourism industry within the cross-border institutional and economic context of the Guangdong-Hong Kong-Macao Greater Bay Area (GBA)? (2) How do these factors interact dynamically, and what are their relative impacts under the unique conditions of the GBA, including differentiated institutional systems, policy coordination, and regional resource endowments? (3) Based on empirical analysis, what context-specific policy and practical recommendations can be proposed to advance the high-quality and sustainable integration of the sports tourism industry in the GBA? By answering these questions, the study extends existing research by situating the analysis within a distinctive cross-border context, thereby bridging the gap in micro-level investigations of sports tourism industry integration. The theoretical contribution lies in constructing a multidimensional mechanism model encompassing government behavior, resource environment, enterprise development, market demand, technological innovation, and human capital, while revealing the systemic and region-specific logic of industry integration. The practical contribution is to provide tailored guidance for policymakers and industry stakeholders in the GBA, supporting more effective strategies to promote deep integration and sustainable development of the sports tourism industry.

This study takes the Guangdong-Hong Kong-Macao Greater Bay Area as a case study, collecting data through enterprise surveys and questionnaires, and employing Structural Equation Modeling (SEM) to empirically analyze the action paths of six key driving factors. The paper is structured as follows: Section 2 outlines the research design, which incorporates interviews and grounded theory to identify and construct a theoretical model of the driving factors in sports tourism industry integration; Section 3 presents the empirical research, detailing data collection through surveys and the application of SEM to assess the action paths and impact intensity of each influencing factor; Section 4 discusses the findings, compares them with existing research, summarizes the key insights and contributions of the study, and identifies remaining issues and avenues for future research; Section 5 concludes with policy recommendations and practical strategies aimed at advancing the deep integration of the sports tourism industry in the GBA, based on the empirical findings of the study.

## Materials and methods

### Research design

This study combines exploratory theoretical construction with empirical testing to analyze the complex mechanisms driving the integrated development of the sports tourism industry in the GBA. The research framework is structured in two phases:

(1) Theoretical construction phase. In the first phase, Grounded Theory methodology is employed to inductively identify the key driving factors for sports tourism industry integration based on qualitative data. This phase involves the collection of in-depth interview data from enterprise representatives and industry association leaders within the GBA sports tourism sector. The data are carefully coded using open coding, axial coding, and selective coding to extract key categories and core driving factors. Additionally, relevant domestic and international literature is reviewed to complement the data and further refine the identified factors. These factors emerge directly from the data without preconceived theoretical frameworks, emphasizing a data-driven approach to theory development. The coding process iteratively refines and clarifies these factors, ensuring that they are grounded in empirical evidence.

(2) Empirical testing phase. In the second phase, the findings from the Grounded Theory coding process are integrated with relevant literature to inform the development of a survey questionnaire. This questionnaire, titled "Guangdong-Hong Kong-Macao Greater Bay Area Sports Tourism Industry Integration Development Influencing Factors Survey," is designed to collect quantitative data for hypothesis testing. The Structural Equation Modeling (SEM) method is then

applied to analyze the survey data and test the proposed relationships among the key driving factors identified in the theoretical phase. While Grounded Theory enables the identification of these core factors, SEM is used to empirically validate the relationships between them, allowing us to test how these factors influence sports tourism industry integration in a quantitative manner. This combined approach ensures both theoretical grounding and empirical verification, enhancing the robustness of our findings.

## Methodological justification

The combination of grounded theory and SEM in this study is justified as follows: grounded theory was first applied to explore the complex and under-researched phenomenon of sports tourism integration in cross-border regions, allowing for the emergence of key factors and relationships from the data. This inductive approach provided a rich understanding of the integration process and identified the key driving factors. Subsequently, SEM was employed to test the relationships between these factors and the integration outcomes, providing a deductive validation of the emerging theory. This mixed-methods approach not only aligns with the iterative nature of social science research but also strengthens the robustness of the findings by combining the depth of qualitative insights with the generalizability of quantitative analysis. The transition from qualitative coding to quantitative model testing was ensured through meticulous alignment between the categories derived from grounded theory and the variables used in the SEM. Specifically, the interviewees' responses were coded into distinct categories, which were then transformed into measurable variables for the questionnaire design, ensuring that the quantitative model accurately reflected the qualitative insights.

## Data collection

Data for this study were collected through two main methods: in-depth interviews and questionnaires.

(1) In-depth Interviews. Interviews are a crucial component of Grounded theory research. In this study, 16 interviewees were purposefully selected from sports tourism-related enterprises and industry associations within the GBA. The interviewees included managers and frontline employees from various sports tourism organizations. Semi-structured in-depth interviews were conducted over a period of six months (August 2023 – March 2024), with each interview lasting between 30–60 minutes. The interviews focused on three core topics: the current status of sports tourism industry integration, the factors influencing integration, and the challenges associated with collaboration. The data were transcribed and analyzed using professional software, producing over 100,000 words of interview material. The interview information is shown in Table 1. The sample size for the in-depth interviews was determined based on the principle of thematic saturation, a core tenet in qualitative and grounded theory research [22,23]. Thematic saturation is achieved when additional interviews no longer yield new thematic insights or conceptual categories, indicating that a comprehensive understanding of the research phenomenon has been reached. After conducting 16 interviews, we performed a preliminary analysis and found that the key themes and categories related to the drivers and mechanisms of sports tourism integration were consistently recurring, suggesting that saturation had been achieved. This approach prioritizes data depth and richness over statistical representativeness, which is well-suited for exploratory theory-building. Furthermore, the 16 interviewees were purposively selected to ensure a diverse representation of roles (e.g., general managers, project planners, association secretaries) and enterprise types across all 11 cities within the GBA, thereby enhancing the contextual comprehensiveness and validity of our qualitative findings.

(2) Questionnaire Survey. In the empirical phase, a structured questionnaire was used to collect quantitative data. A total of 43 initial measurement items were developed, focusing on the factors of integration (independent variables) and the effects of integration (dependent variables). After expert feedback and multiple rounds of refinement, a final questionnaire with 38 measurement items was used. The formal questionnaire is divided into three sections: First,

**Table 1. Interview information.**

| No. | Interviewee Information | Main Business | Interview Duration | Interview Content (Partial Examples) |
|---|---|---|---|---|
| 1 | General Manager of Sports Tourism Destination | Sports tourism, sports fitness and entertainment | 32 minutes | 1. What sports or tourism industries does your enterprise engage in? What is the industrial benefit? |
| 2 | General Manager of Sports Event Operations | Sports event tourism, sports training | 58 minutes | 2. What are the ways or products for the integration of sports and tourism industries in your enterprise? |
| 3 | Secretary-General of Sports Industry Association | Sports industry policy promotion and execution, sports event introduction | 35 minutes | 3. What is the biggest challenge in the integration of sports and tourism industries in your enterprise? |
| 4 | General Manager of Sports Tourism Research and Learning | Development of sports research tourism products | 36 minutes | 4. What are the driving factors for your enterprise's integration of sports tourism? |
| 5 | General Manager of Sports Event Operations | Sailing event operations | 32 minutes | 5. Has the government played any role in promoting your enterprise's integration of sports tourism? How effective is it? |
| 6 | Manager of Sports Event Operations | Sports event operations, cultural and tourism resource integration | 30 minutes | 6. Can the integration of sports and tourism increase the competitiveness of your enterprise and promote transformation? What are the operational benefits after integration? |
| 7 | Sports Tourism Project Planning Manager | Planning of sports tourism team-building projects | 40 minutes | 7. What do you think about the overall innovation atmosphere of the sports tourism industry in GBA? Does the regional innovation environment promote the integration of sports tourism industries in enterprises? |
| 8 | Expert in Cultural, Sports and Tourism Industry Planning | Cultural, sports, and tourism industry planning and management | 42 minutes | 8. Has your enterprise collaborated with other cities in GBA on sports tourism integration? |
| 9 | General Manager of Sports Event Operations | Marathon, basketball, and other event operations | 37 minutes | |
| 10 | Project Manager of Sports Research and Learning Travel | Design of sports tourism routes | 52 minutes | |
| 11 | Product Manager of Cultural, Sports and Tourism Resource Development | Resource integration and product development | 39 minutes | |
| 12 | Manager of Outdoor Sports Club | Organization and planning of outdoor activities | 43 minutes | |
| 13 | Product Manager of Sports Travel Agency | Sports tourism products, route planning | 36 minutes | |
| 14 | Sports Town Operations Specialist | Planning and execution of sports events and sports tourism activities | 25 minutes | |
| 15 | Market Development Specialist of Sports Industry Promotion Association | Market research, brand building, and promotion of the sports industry | 35 minutes | |
| 16 | General Manager of Sports Event Tourism | Sports training, tourism route design | 32 minutes | |

Basic information of respondents. This section includes information about the respondent's city, enterprise size, industry type, and the enterprise's goals regarding sports tourism industry integration. This part helps to understand the respondents' basic background and their level of awareness of the influencing factors in sports tourism industry integration, thereby controlling the quality of the questionnaire responses to some extent. Second, Impact of integration factors on industry development. This section measures the impact of six factors-government support, resource environment, enterprise development, market demand, technological innovation, and human capital on the sports tourism industry integration in GBA. A total of 24 variables are included. Third, Integration development outcomes. This section focuses on the effects and issues associated with integration development, with 14 measurement items. The Likert 7-point scale is used for scoring, with higher scores indicating a greater impact of the variable on the sports tourism industry integration in GBA.

The survey targets for this study included managers, entrepreneurs, and relevant staff members within the sports tourism sector of the GBA. To ensure the representativeness and generalizability of the sample, the selection followed the principles of typicality, comprehensiveness, and inter-group differences. The research area covered 11 cities: Shenzhen,

Guangzhou, Hong Kong, Macao, Foshan, Dongguan, Huizhou, Zhuhai, Jiangmen, Zhaoqing, and Zhongshan. The questionnaire was distributed through the "Wenjuanxing" online platform and was made available from August 1 to August 12, 2023, to the WeChat groups of professionals in the sports industry and tourism industry within the Guangdong-Hong Kong-Macao Greater Bay Area. All participants were provided with an informed consent form through the survey cover letter before completing the questionnaire. A total of 1,500 questionnaires were distributed, with 1,500 responses collected. After excluding invalid responses, 1,468 valid questionnaires were obtained, resulting in a valid response rate of 97.8% (Table 2). To ensure the reliability of the sample data, common method bias and non-response bias were assessed. Using the Harman single-factor test, it was found that the variance explained by the first principal component was 38.27%, which is below the 50% threshold. The seven extracted principal components together explained 73.24% of the total variance, suggesting no significant common method bias. Additionally, confirmatory factor analysis (CFA) using AMOS 24.0 software revealed that the fit of the single-factor model was worse than that of the multi-factor model with seven variables, further confirming that common method bias is not an issue. For non-response bias, the returned questionnaires were divided into the first 10% and last 10% based on response time, and a two-tailed test was performed on respondents' basic information and enterprise data in both groups. The results showed no significant differences ($p > 0.05$), indicating no substantial non-response bias.

**Table 2. Descriptive statistics of survey sample.**

| Basic Information | Option | Frequency | Percentage (%) | Valid Percentage (%) | Cumulative Percentage (%) |
|---|---|---|---|---|---|
| Location of the Company | Guangzhou | 369 | 25.1 | 25.1 | 25.1 |
| | Shenzhen | 393 | 26.8 | 26.8 | 51.9 |
| | Foshan | 117 | 8 | 8 | 59.9 |
| | Dongguan | 132 | 9 | 9 | 68.9 |
| | Huizhou | 84 | 5.7 | 5.7 | 74.6 |
| | Zhaoqing | 59 | 4 | 4 | 78.6 |
| | Zhuhai | 61 | 4.2 | 4.2 | 82.8 |
| | Zhongshan | 71 | 4.8 | 4.8 | 87.6 |
| | Jiangmen | 63 | 4.3 | 4.3 | 91.9 |
| | Hong Kong | 59 | 4 | 4 | 95.9 |
| | Macao | 60 | 4.1 | 4.1 | 100 |
| | Total | 1468 | 100 | 100 | |
| Company Size | Large Enterprise (Employees > 500) | 335 | 22.8 | 22.8 | 22.8 |
| | Medium Enterprise (Employees 100–500) | 629 | 42.8 | 42.8 | 65.6 |
| | Small Enterprise (Employees 100−10) | 389 | 26.4 | 26.4 | 92 |
| | Micro Enterprise (Employees < 10) | 115 | 7.8 | 7.8 | 100 |
| | Total | 1468 | 100 | 100 | |
| Industry Type | Sports Industry | 333 | 22.7 | 22.7 | 100 |
| | Entertainment Industry | 346 | 23.6 | 23.6 | 100 |
| | Tourism Industry | 500 | 34.1 | 34.1 | 100 |
| | Training Industry | 376 | 25.6 | 25.6 | 100 |
| | Health Industry | 209 | 14.2 | 14.2 | 100 |
| | Equipment Industry | 172 | 11.7 | 11.7 | 100 |
| | Other | 386 | 26.3 | 26.3 | 100 |

This study was approved by Medical Ethics Committee of the Medical School of Shenzhen University. Before participation, all respondents were provided with clear information regarding the purpose of the study, the voluntary nature of their participation, and the confidentiality of their responses. Informed written consent was obtained from all participants prior to their completion of the online questionnaire. Participants explicitly agreed to participate by clicking a consent button on the "Wenjuanxing" online platform, which served as their written consent. The research strictly adhered to the ethical guidelines for human subjects research, ensuring participant anonymity and data privacy.

## Factor identification

**Open coding.** Open coding is a systematic approach designed to analyze raw data and, through an iterative process of revision, validation, and refinement, extract and construct core concepts and category systems. This process is divided into three steps: labeling, conceptualizing, and categorizing. The first step is labeling, where the raw data is analyzed sentence by sentence, and preliminary refinements are made based on the interview guide, resulting in short phrases. The second step is conceptualization, which involves further refining the commonalities between the labels already created to condense the material. In this study, 30 concepts were formed during this step. The final step is categorization. After labeling and conceptualization, the organization and clarity of the raw data improved significantly. However, the 30 concepts formed were still too extensive, making it difficult to focus on the core issues of the research. Therefore, further extraction of related concepts from the set of labels was necessary. Based on the 30 concepts, this study distilled 13 initial categories [24–41], as shown in S1 Table. Given that each concept entry corresponds to a wealth of raw data codes, for simplicity and clarity, this paper only selects one representative raw statement and its corresponding initial concept for illustration. Each category was refined to capture the core elements driving sports tourism industry integration.

**Axial coding.** Axial coding is a process of analyzing and integrating the initial categories by clarifying the logical sequence and relationships between the categories, further refining them into higher-level core categories. In this study, the 13 initial categories were deeply analyzed and categorized, ultimately leading to the identification of 6 core categories: government behavior, resource environment, enterprise development, market demand, technological innovation, and human capital. The results of axial coding are shown in S2 Table.

**Selective coding.** Selective coding synthesizes the relationships among the six core categories into a unifying "core category." This core category, identified as "Driving Factors and Mechanisms of Sports Tourism Industry Integration," integrates the key factors derived from the qualitative data. The "storyline" summarizes the relationships between these factors and divides them into three broad categories: (1) Resource Environment as the foundation condition: This factor represents the fundamental environmental context that supports the integration of sports tourism. It includes resource endowment (such as geographical and natural resources) and the innovative environment (such as infrastructure and technological support). These elements lay the groundwork for the development of sports tourism and provide the necessary physical and technological conditions for industry integration. (2) Government Behavior and Market Demand as external conditions: These are the external factors that influence the integration process. Government behavior refers to the policies, fiscal policies, and financial incentives that either facilitate or hinder the integration of the sports tourism industry. Market demand reflects the consumer demand and market competition that shape industry trends, push innovation, and drive the evolution of sports tourism offerings. These external factors shape the operational environment for enterprises and influence the broader market dynamics. (3) Enterprise Development Needs, Technological Innovation, and Human Capital as endogenous drivers: These are the internal dynamics that stimulate and sustain the growth of the sports tourism industry. Enterprise development needs refer to the internal performance and upgrading requirements of firms, driving them to innovate and enhance their capabilities. Technological innovation refers to the application of new technologies and innovation-driven strategies that improve service quality, increase operational efficiency, and foster industry synergies. Human capital, including innovative talent and employee cognition, plays a vital role in improving the

quality of services and products, and ensuring the industry's competitiveness and sustainable growth. These categories outline the mechanisms through which the factors contribute to industry integration.

## Model construction

Based on the identified categories and relationships, a conceptual model is developed to represent the dynamic mechanisms driving sports tourism industry integration (Fig 2). The model begins with six key factors: government behavior, resource environment, enterprise development needs, market demand, technological innovation, and human capital. It systematically reveals the pathways through which each factor influences the integration effect of the sports tourism industry. The research hypotheses proposed are derived from the grounded theory analysis, which provides a theoretical foundation for the SEM testing. The hypotheses are as follows:

H1: Government behavior has a significant positive impact on the integration effect of the sports tourism industry.

H2: Resource environment has a significant positive impact on the integration effect of the sports tourism industry.

H3: Enterprise development needs have a significant positive impact on the integration effect of the sports tourism industry.

H4: Market demand has a significant positive impact on the integration effect of the sports tourism industry.

H5: Technological innovation has a significant positive impact on the integration effect of the sports tourism industry.

H6: Human capital has a significant positive moderating effect on the integration effect of the sports tourism industry.

## Variable definition

To achieve the research objectives and improve the quality of the questionnaire, this study avoids using a single indicator to measure the research variables. A single indicator is insufficient to fully reflect the essence of the variable, which may lead to partial results. Therefore, a multi-indicator approach is used to measure each variable. The selection of indicators is based on scientific and objective principles, considering the following two aspects: first, existing research literature is referenced to systematically review relevant studies and determine appropriate indicators; second, the practical situation of sports tourism industry integration in GBA is considered, with adjustments and additions to the indicators based on information gathered from preliminary research and in-depth interviews. Based on the above methodology, this study decomposes the research variables into multiple indicators, using a comprehensive and averaging approach to mitigate the measurement errors that might arise from a single indicator (Table 3). This multi-indicator design enhances the reliability of the variable measurements and strengthens the scientific and authoritative nature of the scale.

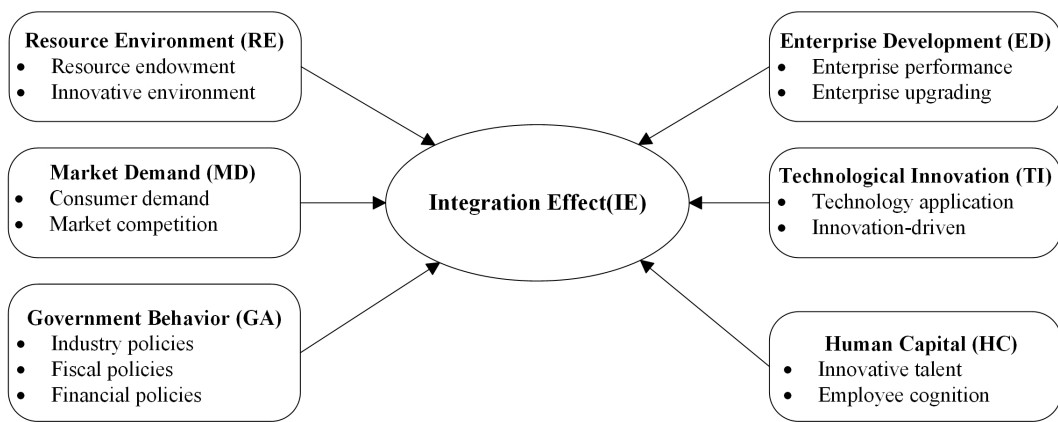

**Fig 2. Conceptual model of the dynamic mechanisms of sports tourism industry integration.**

**Table 3. Variables and item codes.**

| Variable | Item | Code |
|---|---|---|
| Government Behavior (GA) | Support for sports tourism industry through fiscal funding | GA1 |
| | Land policy for sports tourism industry | GA2 |
| | Loan policy for sports tourism industry | GA3 |
| | Guidance from local governments for sports tourism industry | GA4 |
| | Reward/subsidy policies for the sports tourism industry | GA5 |
| | Government guidance on public participation in sports tourism consumption | GA6 |
| Resource Environment (RE) | Local sports/tourism resource endowment | RE1 |
| | Sports/tourism infrastructure conditions | RE2 |
| | Local transportation and location conditions | RE3 |
| | Local innovation environment conditions | RE4 |
| Enterprise Development (ED) | Development demand for improving business efficiency | ED1 |
| | Development demand for enterprise transformation and upgrading | ED2 |
| | Development demand for enhancing enterprise competitiveness | ED3 |
| | Leadership role of leading enterprises | ED4 |
| Market Demand (MD) | Consumer demand for experiential tourism | MD1 |
| | Market competition in sports/tourism products | MD2 |
| Technological Innovation (TI) | Promotion of sports technology | TI1 |
| | Adoption of intelligent sports tourism services | TI2 |
| | Application of new marketing technologies | TI3 |
| | Construction of sports industry park bases | TI4 |
| | Innovation in integration methods between sports and related industries | TI5 |
| Human Capital (HC) | Innovative talent in enterprises | HC1 |
| | Innovation capability training for enterprise employees | HC2 |
| | Employee cognition of new business models | HC3 |
| Integration Effect (IE) | Increase in enterprise revenue in recent years | IE1 |
| | Significant enhancement of enterprise competitiveness in recent years | IE2 |
| | Rapid extension of the enterprise industry chain in recent years | IE3 |
| | Rapid expansion of enterprise customer base in recent years | IE4 |
| | Rapid increase in enterprise brand recognition in recent years | IE5 |
| | Increase in cross-industry integration in recent years | IE6 |

## Results

To empirically validate the conceptual model and hypotheses (H1-H6) derived from the grounded theory analysis, Structural Equation Modeling (SEM) was employed using the survey data. The findings regarding the hypothesized paths are presented below.

### Reliability and validity testing

This study conducted a detailed examination of the reliability and validity of the scale using SPSS 26.0 and AMOS 24.0, with the results presented in Table 4. In terms of reliability, the Cronbach's α values for all variables exceeded 0.7, indicating good internal consistency and stability. To further validate the scale's validity, Confirmatory Factor Analysis (CFA) was performed. The results revealed satisfactory convergent validity (CR > 0.70, AVE > 0.50) and discriminant validity. Factor loadings for all measurement items ranged between 0.849 and 0.893, exceeding the 0.7 threshold and confirming strong

**Table 4. Reliability and validity test results.**

| Variable | Items | Cronbach's α | Factor Loadings Range | CR | AVE | KMO |
|---|---|---|---|---|---|---|
| Government Behavior (GA) | 6 | 0.951 | 0.849–0.893 | 0.896 | 0.679 | 0.934 |
| Resource Environment (RE) | 4 | 0.928 | 0.851–0.886 | 0.912 | 0.728 | 0.860 |
| Enterprise Development (ED) | 4 | 0.928 | 0.856–0.885 | 0.922 | 0.735 | 0.860 |
| Market Demand (MD) | 2 | 0.733 | 0.854–0.864 | 0.943 | 0.762 | 0.635 |
| Technological Innovation (TI) | 5 | 0.940 | 0.863–0.882 | 0.909 | 0.684 | 0.911 |
| Human Capital (HC) | 3 | 0.908 | 0.859–0.892 | 0.932 | 0.748 | 0.753 |
| Integration Effect (IE) | 6 | 0.952 | 0.868–0.896 | 0.917 | 0.716 | 0.941 |

explanatory power of latent constructs. The Kaiser-Meyer-Olkin (KMO) values (0.635–0.941) indicated suitability for factor analysis. In summary, the scale demonstrated excellent robustness for empirical analysis.

## Structural model goodness of fit test

As shown in Table 5, the fit indices of the structural equation model for the effects of sports tourism industry integration in GBA meet or exceed the recommended threshold values, indicating that the model fits the data well. Specifically, the chi-square to degrees of freedom ratio (CMIN/DF) is 1.857, which is significantly lower than the recommended value of 3, with a p-value of 0.052, which is close to the significance level, suggesting a good model fit. Further fit indices show that the Normed Fit Index (NFI), Comparative Fit Index (CFI), Goodness of Fit Index (GFI), and Relative Fit Index (RFI) of the model are 0.989, 0.995, 0.975, and 0.985, respectively, all of which are well above the recommended threshold of 0.9, demonstrating high fit and improvement of the model. Additionally, the Root Mean Square Residual (RMR) and Root Mean Square Error of Approximation (RMSEA) are 0.015 and 0.024, respectively, both of which are below the recommended value of 0.05, indicating small residuals and good model parsimony. These results collectively demonstrate strong data-theory alignment, minimal residuals, and high model parsimony, fulfilling SEM fit criteria (Hu & Bentler, 1999). The model thus robustly captures the driving mechanisms of sports tourism integration in the GBA, providing reliable theoretical and practical foundations.

## Hypothesis testing

This study employed Structural Equation Modeling (SEM) to empirically test the research hypotheses proposed earlier. The results showed that the Structural Equation Modeling validated all hypotheses (H1-H6), confirmed the positive impacts of six drivers on sports tourism integration (p < 0.05), but there were significant differences in the intensity of the effects (Table 6).

Hypothesis H1: Consistent with hypothesis H1, government behavior (GA) shows a statistically significant positive effect on integration outcomes (β = 0.052, p = 0.002), supporting its role as an external facilitator identified in the theoretical framework. However, the modest effect size suggests limitations in policy reach and implementation efficiency within the cross-border context, aligning with qualitative insights about policy fragmentation.

Hypothesis H2: Resource environment (RE) as a driver receives partial support (β = 0.041, p = 0.017). While statistically significant, the minimal effect size indicates that the resource endowment and innovation environment, though

**Table 5. Actual and reference values of structural equation model fit indices.**

| | CMIN/DF | P | NFI | CFI | GFI | RFI | RMR | RMSEA |
|---|---|---|---|---|---|---|---|---|
| Actual Value | 1.857 | 0.052 | 0.989 | 0.995 | 0.975 | 0.985 | 0.015 | 0.024 |
| Reference Value | <3 | >0.05 | >0.9 | >0.9 | >0.9 | >0.9 | <0.05 | <0.05 |

**Table 6. Structural equation estimation results.**

| Hypothesis | Path | Standardized coefficients (β) | p-value | Support |
|---|---|---|---|---|
| H1 | GA→IE | 0.052 | 0.002(**) | Yes |
| H2 | RE→IE | 0.041 | 0.017 (*) | Yes |
| H3 | ED→IE | 0.321 | <0.001 (***) | Yes |
| H4 | MD→IE | 0.065 | <0.001 (***) | Yes |
| H5 | TI→IE | 0.541 | <0.001 (***) | Yes |
| H6 | HC→IE | 0.619 | <0.001 (***) | Yes |

<sup>a</sup>Significance: *p < 0.05, **p < 0.01, ***p < 0.001.

foundational, have limited direct impact on integration at the micro-enterprise level, which could reflect the varied resource availability and inefficiencies in resource distribution across the GBA.

Hypothesis H3: Enterprise development (ED) showed moderate influence (β = 0.321, p < 0.001). This indicates that internal efficiency and upgrading demands within enterprises serve as moderate yet meaningful catalysts for sports tourism integration.

Hypothesis H4: Market demand (MD) show a statistically significant but limited direct impact (β = 0.065, p < 0.001), reflecting the early developmental stage of the sports tourism market in the GBA, where consumer preferences and demand patterns are still evolving rapidly.

Hypothesis H5: Technological innovation was found to have a strong positive effect (β = 0.541, p < 0.001), This underscores the crucial role of new technologies and innovation capabilities in enhancing competitiveness and operational efficiency within the sports tourism industry, as emphasized in the theoretical framework.

Hypothesis H6: Hypothesis H6, positing human capital (HC) as a driver, receives strong support (β = 0.619, p < 0.001). This robust finding empirically confirms the centrality of innovative talent and employee cognition highlighted in the grounded theory analysis as the paramount driver of integration success.

The structural model (Fig 3) further clarifies these relationships, confirming that HC and TI constitute the core endogenous forces propelling integration, while policy and resource factors require targeted refinement. Collectively, these findings provide robust empirical support for strategic interventions toward sustainable sports tourism integration in the GBA.

## Discussion

The empirical findings provide robust quantitative validation for the multidimensional drivers of sports tourism integration identified through the grounded theory approach. The SEM results largely confirm the proposed conceptual model, revealing significant positive effects for all six hypothesized drivers (H1-H6 supported). Crucially, the analysis elucidates the relative strength of these drivers, with human capital (HC) and technological innovation (TI) emerging as the dominant endogenous forces (β = 0.619 and β = 0.541, respectively), while government behavior (GA) and resource environment (RE), though significant, exhibit considerably weaker direct impacts (β = 0.052 and β = 0.041). Through the quantitative verification of the dynamic mechanism of the integration of the sports tourism industry proposed by the qualitative model, the our theoretical framework has been strengthened.

Existing literature has generally concentrated on macro-level factors influencing sports tourism industry integration, such as policy support, technological advancements, and market demand. However, these studies predominantly focus on interpreting policy documents and regional economic data analysis [16,21], often overlooking micro-level elements such as enterprise behavior, technological applications, and human capital. In contrast, this study's distinctive contribution lies in its focus on the Greater Bay Area (GBA), a representative cross-border region. By systematically examining the roles and impact pathways of six key drivers—government behavior, resource environment, enterprise development, market

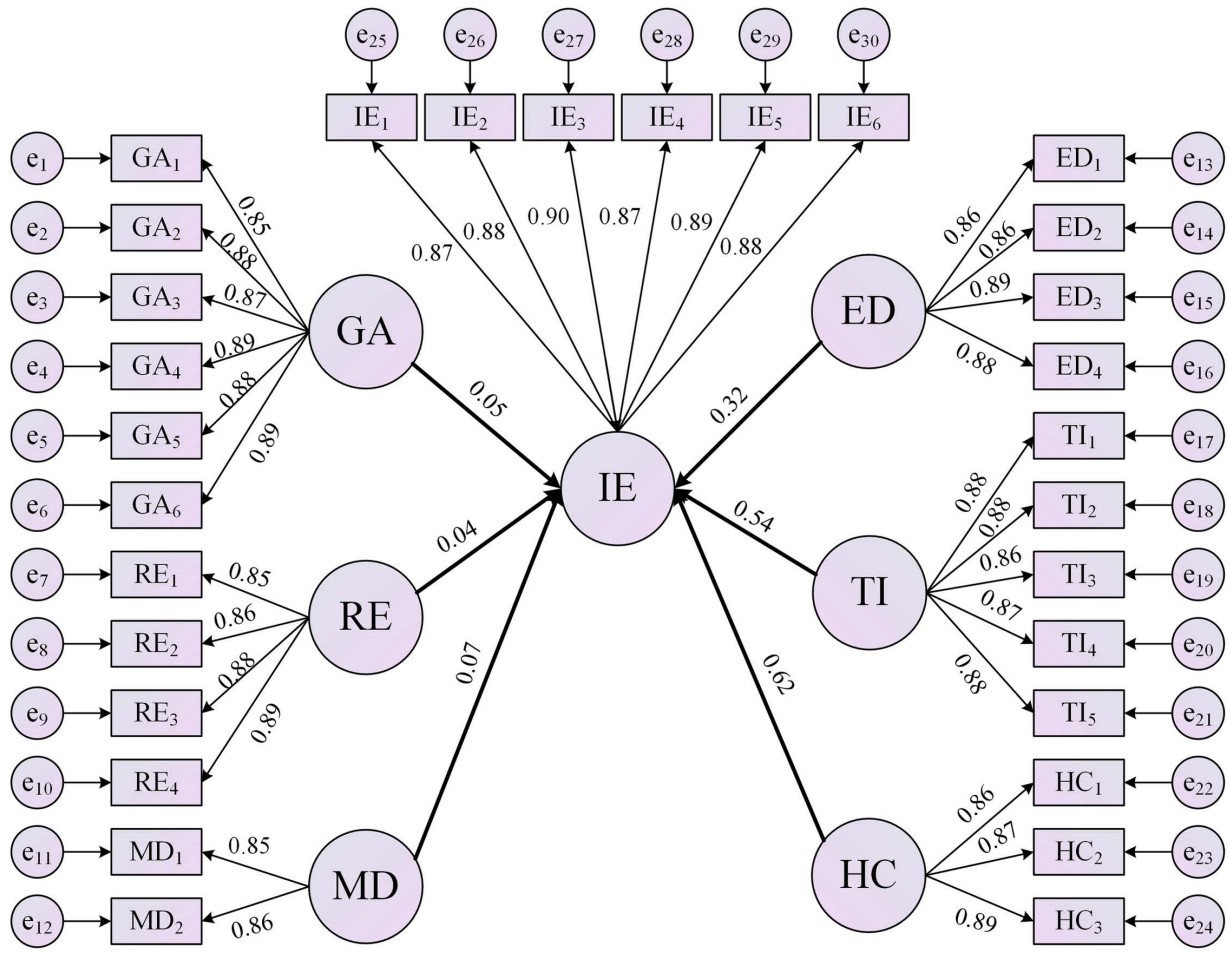

**Fig 3. Structural equation model of driving factors for sports tourism industry integration.**

demand, technological innovation, and human capital—this research provides new empirical insights into the micro-level forces shaping sports tourism integration. The findings underscore that human capital and technological innovation are the core drivers of industry integration, in alignment with the existing literature on innovation-driven development, but they also emphasize the critical role of human capital at the micro level—a dimension that has received limited attention in previous research.

While government behavior (GA) and resource environment (RE) exert some influence on sports tourism industry integration, their impact is notably weaker in this study. The role of government policy in the GBA primarily manifests in fiscal support and policy guidance. However, the empirical results show that the practical effects of these policies are mitigated by challenges such as limited policy coverage and depth, particularly with respect to cross-regional coordination. The weak β values for GA (0.052) and RE (0.041) in this study align with theoretical insights from the grounded theory analysis, where the concepts of 'policy fragmentation' and 'inefficient resource allocation' were identified as key barriers. These challenges are particularly pronounced in cross-border regions like the GBA, where coordination difficulties and disparities in policy enforcement between regions hamper the effectiveness of governmental interventions. As a result, while government actions provide foundational support for integration, their influence at the micro-enterprise level remains suboptimal, thereby not reaching the expected intensity as hypothesized.

Similarly, while the resource environment (RE) contributes essential public resources for regional industry develop-ment, its direct effect at the micro-level is modest. This highlights the need for more efficient distribution and targeted allocation of resources. The resource environment in the GBA has not fully addressed the personalized needs of different enterprises, thus constraining its ability to propel industry integration. However, it is important to note that even with their relatively low direct impacts, government policies and resource environments remain essential for shaping the broader landscape of sports tourism development. Government policies create necessary infrastructure, while the resource envi-ronment provides the foundational public goods required for industry growth.

When comparing this study with research conducted in other regions (such as the Yangtze River Delta and Beijing-Tianjin-Hebei), some unique features emerge. In these other regions, the integration of the sports tourism industry typi-cally benefits from stronger policy support and more efficient cross-regional coordination and resource allocation [17,42]. In contrast, the GBA faces considerable policy execution bottlenecks and resource allocation inefficiencies, which limit the influence of government behavior and resource environment on the integration process. This distinction underscores that while the GBA holds significant strategic potential for sports tourism industry integration, enhancing the effectiveness of its policies and resource allocation remains a critical step for maximizing this potential.

The integration process for the sports tourism industry is an ongoing, and factors such as the policy environment, technological progress, and market demand may exert different effects at various stages. While this study is based on cross-sectional data, it does not account for how changes in policies, technological advances, and other factors might influence the dynamic evolution of industry integration over time. Future research could expand the scope of this study by incorporating longitudinal data and case tracking, which would allow for a deeper understanding of the long-term impacts of policy changes and technological developments on industry integration.

Moreover, while the GBA serves as a valuable and representative case for cross-border regions, its specific policy con-text, economic development levels, and market demand patterns may differ from those in other regions. Thus, the gen-eralizability of this study's conclusions to other regions requires further examination. Future research could enhance the external validity of these findings by broadening the sample scope and conducting comparative analyses across different cross-border regions. Additionally, future studies could explore mechanisms for strengthening the impact of government behavior and resource environment on sports tourism industry integration, such as through optimizing policy implementa-tion frameworks and improving the efficiency of resource allocation.

## Conclusions and recommendations

### Main conclusions

This study set out to explore the integration dynamics of the sports tourism industry within the distinctive cross-border con-text of the Guangdong-Hong Kong-Macao Greater Bay Area (GBA). Guided by the research questions, the findings reveal that government behavior, resource environment, enterprise development, market demand, technological innovation, and human capital constitute the core driving factors of integration, but their mechanisms and impact intensities vary in ways that reflect the region's unique characteristics of multi-jurisdictional governance, cultural diversity, and resource comple-mentarity. By empirically testing these relationships through enterprise survey data and Structural Equation Modeling (SEM), the study contributes a context-specific and evidence-based understanding of how integration unfolds in the GBA.

The findings reveal that human capital and technological innovation stand out as the most powerful endogenous drivers, highlighting the critical role of the GBA's vast talent pool and digital adoption capabilities in overcoming inherent cross-border barriers. Enterprise development needs further emerged as a strong internal catalyst, underscoring the pro-active role of leading firms in driving cross-border collaboration and industrial transformation. In contrast, market demand, resource environment, and government behavior—though statistically significant—exerted limited direct influence. This pattern underscores the constraining effects of market segmentation, resource allocation inefficiencies, and most notably, policy fragmentation resulting from the "one country, two systems" framework.

 

These results emphasize that the dynamics of sports tourism integration in the GBA are deeply shaped by its unique cross-border governance structure, socio-economic diversity, and institutional landscape. The originality of this study lies in its ability to bridge micro-level dynamics—such as enterprise behaviors and talent mobility—with macro-level institutional frameworks, offering a finely-grained, context-sensitive explanation of integration mechanisms that is firmly grounded in empirical evidence.

The upcoming 15th National Games presents a strategic opportunity to translate these findings into practice. Efforts should focus on enhancing policy coordination, building digital resource-sharing platforms, developing cross-border tourism products, and facilitating talent mobility. While the study is based on cross-sectional data from the GBA, its theoretical framework and methodological approach offer valuable insights for research in other cross-border regions. Future studies should adopt longitudinal designs and extend comparative analyses to other bay areas or multi-jurisdictional settings to further refine the dynamic mechanisms of regional sports tourism integration.

## Recommendations

Building upon the empirical findings and the distinctive cross-border context of the Guangdong-Hong Kong-Macao Greater Bay Area (GBA), the following recommendations are proposed to promote the high-quality integration of the sports tourism industry:

(1) Enhance cross-border policy coordination and institutional alignment: Given the constraints of fragmented governance under the "one country, two systems" framework, governments at different levels should strengthen coordination mechanisms, expand the scope of targeted subsidies, and streamline cross-border administrative procedures. Efforts should focus on aligning land-use approvals, fiscal incentives, and cooperation frameworks across Guangdong, Hong Kong, and Macao, thereby reducing policy-induced barriers for enterprises and facilitating a more integrated sports tourism market.

(2) Leverage technological innovation to build a digital integration platform: Technological innovation is a dominant endogenous driver in the GBA. It is essential to develop digital platforms that integrate big data, cloud computing, and artificial intelligence to enable resource sharing, visitor flow monitoring, and personalized product design across jurisdictions. A region-wide "Sports Tourism Digital Hub" could enhance efficiency and connectivity, while also providing data-driven insights for policymakers and enterprises to refine strategies.

(3) Strengthen enterprise-led collaboration and cross-border industrial clusters: The proactive role of leading enterprises in the GBA highlights the need to support enterprise-driven collaborations. Governments should encourage the formation of cross-border industrial clusters in sports event tourism, outdoor experiences, and wellness tourism, with flagship enterprises serving as anchors. Public-private partnerships (PPP) should be promoted to improve investment efficiency and create synergies across cities, reducing redundant projects and fostering regional complementarities.

(4) Cultivate and mobilize interdisciplinary talent across borders: Human capital is the strongest driver of integration. Talent exchange platforms should be established to facilitate the free flow of sports tourism professionals, researchers, and entrepreneurs across Guangdong, Hong Kong, and Macao. Industry-academia collaboration programs and cross-border training initiatives can provide continuous talent supply. Incentives such as tax benefits and innovation grants should be introduced to attract high-end professionals, reinforcing the GBA's position as a global talent hub for sports tourism.

(5) Expand and differentiate segmented markets through regional branding: While market demand shows limited direct effects, its segmented potential should be actively cultivated. The GBA should capitalize on niche segments such as mega-event tourism (e.g., the 15th National Games), outdoor adventure, and health-focused sports tourism. Joint branding initiatives across Guangdong, Hong Kong, and Macao should be launched to enhance global recognition, positioning the GBA as a world-class destination for cross-border sports tourism.

In sum, these recommendations rooted in the distinctive institutional and socio-economic context of the Guangdong-Hong Kong-Macao Greater Bay Area, provide a roadmap for deepening the integration of the sports tourism industry. By enhancing governance coordination, leveraging digital technologies, fostering enterprise collaboration, cultivating cross-border talent, and expanding segmented markets, the region can strengthen its global competitiveness while advancing sustainable development goals. Beyond their practical relevance for the GBA, these strategies also offer valuable reference points for other cross-border regions seeking to harness sports tourism as a driver of regional cooperation and growth.

## Supporting information

**S1 Table. Example of open coding.**
(DOCX)

**S2 Table. Results of axial coding.**
(DOCX)

## Acknowledgments

We would like to thank the anonymous reviewers for their constructive feedback and detailed suggestions.

## Author contributions

**Conceptualization:** Jun Yuan.

**Data curation:** Jun Yuan, Canyu Chen.

**Formal analysis:** Jun Yuan, Canyu Chen.

**Funding acquisition:** Jun Yuan.

**Investigation:** Canyu Chen, Yanhong Liu.

**Methodology:** Canyu Chen.

**Project administration:** Jun Yuan, Yanhong Liu.

**Resources:** Yanhong Liu.

**Supervision:** Yanhong Liu.

**Writing – original draft:** Jun Yuan, Canyu Chen, Zhouxin Luo.

**Writing – review & editing:** Jun Yuan, Zhouxin Luo.

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
