## [Decision Letter · Decision Letter 0]

9 Apr 2025

Dear Dr. Yuan,

Thank you for submitting your manuscript to PLOS ONE. After careful consideration, we feel that it has merit but does not fully meet PLOS ONE’s publication criteria as it currently stands. Therefore, we invite you to submit a revised version of the manuscript that addresses the points raised during the review process.

We look forward to receiving your revised manuscript.

Kind regards,

A S Sochipem Zimik

Academic Editor

PLOS ONE

[This work was supported by the National Social Science Foundation of China(20BTY054).].

Additional Editor Comments (if provided):

Reviewers' comments:

Reviewer's Responses to Questions

**Comments to the Author**

1. Is the manuscript technically sound, and do the data support the conclusions?

Reviewer #1: Partly

2. Has the statistical analysis been performed appropriately and rigorously?

Reviewer #1: Yes

3. Have the authors made all data underlying the findings in their manuscript fully available?

Reviewer #1: Yes

4. Is the manuscript presented in an intelligible fashion and written in standard English?

Reviewer #1: Yes

Reviewer #1: The manuscript attempts to address the integration dynamics of the sports tourism industry in the GBA using grounded theory and SEM. My comments are as follows:

1. The study’s core contribution mirrors already established frameworks such as ESP (Environment-System-Policy), which has been previously applied in various regional industrial analyses. The claims of novelty are overstated, particularly as most identified drivers (government policy, resource endowment, enterprise behavior, etc.) are well-documented in existing literature.

2. The theoretical application of ESP is not critically developed. The authors claim it integrates “environmental, systemic, and policy” dimensions but fail to operationalize these within the SEM model in a robust way.

3. The model is visually cluttered, with arrows lacking directionality clarity and undefined pathways. No numerical values or standardized paths are shown on the figure itself.

4. The claim of using grounded theory followed by SEM is methodologically inconsistent. Grounded theory typically builds theory from data without preconceived models, while SEM tests hypothesized models. This hybrid approach is not properly justified, and the transition between qualitative coding and quantitative model testing lacks transparency.

5. While the SEM model yields statistical significance for all six hypotheses, the impact of core constructs like market demand (β=0.065) and government behavior (β=0.052) is minimal.

6. Table 8 and 9 contain too much redundant data without corresponding synthesis in the main text. The lack of condensed insights or grouped patterns makes interpretation difficult and offers minimal value to readers.

7. The manuscript contains repetitive phrasing, vague expressions ("significant support", "relatively weak effect", "key driver") and insufficient transitions between theoretical constructs and empirical results.

8. The authors are encouraged to refine their theoretical framing, consider an alternative modeling strategy (e.g., PLS-SEM or system dynamics), and increase the scientific contribution before resubmitting.

**Do you want your identity to be public for this peer review?** For information about this choice, including consent withdrawal, please see our Privacy Policy

Reviewer #1: No

---

## [Author Response · Author response to Decision Letter 1]

8 Jun 2025

Dear reviewer and editor:

Thank you for your comments and suggestion concerning our manuscript. The comments and suggestions are all valuable and very helpful for revising and improving our paper as well as the important guiding significance to our researches. We have studied comments carefully and have made correction which we hope meet with approval.

To facilitate this discussion, we first retype your comments in italic font and then present our responses to the comments.

Comment 1:

The study’s core contribution mirrors already established frameworks such as ESP (Environment-System-Policy), which has been previously applied in various regional industrial analyses. The claims of novelty are overstated, particularly as most identified drivers (government policy, resource endowment, enterprise behavior, etc.) are well-documented in existing literature.

Response 1:

Thank you for your valuable suggestion regarding our research framework. We understand your concerns about the novelty of the ESP framework. In light of your feedback, we have decided to remove the ESP framework and directly identify the six key driving factors using Grounded Theory (GT), while also constructing a dynamic mechanism model. The use of Grounded Theory ensures that the research conclusions emerge naturally from the data, enhancing the scientific rigor and robustness of the study. The specific modifications are as follows:

(1)Adjustment of the Theoretical Framework: The introduction of the ESP framework has been removed, and we now directly employ Grounded Theory as the research methodology. We emphasize the scientific and rigorous nature of Grounded Theory in extracting core driving factors from the data.

(2)Reconstruction of the Theoretical Model: Using the three stages of coding in Grounded Theory (open coding, axial coding, and selective coding), we have built relationships among the driving factors. The focus is on revealing how the six driving factors (government behavior, resource environment, enterprise development, market demand, technological innovation, and human capital) interact to promote the integration of the sports tourism industry.

(3)Enhanced Empirical Analysis: We use Structural Equation Modeling (SEM) to verify the effects of the six driving factors on the outcomes of industry integration.

(4)Revised Contribution Statement: We have revised the contribution section to highlight：Firstly, the identification of the six driving factors and their dynamic mechanisms through Grounded Theory and empirical research. Secondly, the provision of specific policy recommendations for cross-border regions, such as the Greater Bay Area (GBA). Thirdly, the study addresses the gaps in existing research regarding the micro-level factors (enterprise behavior, technological innovation, and human capital).

The above revisions can be found in the sections of “Research design” “Selective coding” “Model construction” and “Discussion” in the text, with changes highlighted in red. We believe these adjustments will strengthen the clarity and focus of the study, and we hope they meet your expectations.

Comment 2:

The theoretical application of ESP is not critically developed. The authors claim it integrates “environmental, systemic, and policy” dimensions but fail to operationalize these within the SEM model in a robust way.

Response 2:

Thank you for your valuable feedback on this study. We fully understand your concern regarding the critical development of the theoretical application of the ESP framework. You pointed out that while the framework claims to integrate “environmental, systemic, and policy” dimensions, it fails to operationalize these within the Structural Equation Modeling (SEM) in a robust manner. In light of your feedback, we have decided to remove the ESP framework and instead directly identify the six key driving factors using Grounded Theory, while also constructing a dynamic mechanism model. This modification refines the focus of the study to data-driven conclusions, enhancing the scientific rigor and robustness of the research.

Additionally, we have employed Structural Equation Modeling (SEM) to validate the impact of the six driving factors on the effectiveness of sports tourism industry integration. The results show that all path coefficients are significant (P < 0.05), indicating that the six driving factors have a positive influence on industry integration outcomes. Specifically, Human Capital (HC) and Technological Innovation (TI) have the most significant impact on industry integration, with standardized path coefficients of 0.619 and 0.541, respectively. This finding underscores the important role of micro-level factors in driving industry integration.

The above revisions can be found in the sections of “Model construction” and “Hypothesis testing” in the text, with changes highlighted in red. We hope this addresses your concern and enhances the clarity of our theoretical approach and empirical analysis.

Comment 3:

The model is visually cluttered, with arrows lacking directionality clarity and undefined pathways. No numerical values or standardized paths are shown on the figure itself.

Response 3:

Thank you for your valuable feedback on the visual presentation of the model. We appreciate your observations regarding the cluttered appearance and lack of clarity in the directional arrows and pathways. In response to your concerns, we have made the following adjustments:

(1)Clarified Directionality and Pathways: We have revised the model to ensure that all arrows are clearly defined and point in the appropriate direction, improving the readability and interpretability of the diagram.

(2)Simplified Visual Layout: The overall layout of the model has been streamlined to reduce visual clutter. We have reorganized the elements in a more structured and straightforward manner to enhance clarity.

(3)Clarification on Standardized Path Coefficients: Since the path coefficients are derived from the empirical analysis (using Structural Equation Modeling, SEM), they are based on data that will be presented later in the study. As the empirical analysis has not yet been conducted at the model construction stage, we believe it is more appropriate to focus on the theoretical relationships between the variables in this diagram. We present the numerical values and standardized path coefficients after the SEM analysis in the results section, ensuring the data-driven nature of the findings, as shown in Figure 3.

(4)Updated Figure: The updated version of the Figure 2 now visually reflects the theoretical model’s structure, showing the relationships between the key factors without prematurely including the standardized path coefficients. This change improves the clarity and correctness of the figure.

The above revisions can be found in the sections of “Model construction” and “Hypothesis testing” in the text, with changes highlighted in red. We hope these revisions meet your expectations and improve the presentation of our study.

Comment 4:

The claim of using grounded theory followed by SEM is methodologically inconsistent. Grounded theory typically builds theory from data without preconceived models, while SEM tests hypothesized models. This hybrid approach is not properly justified, and the transition between qualitative coding and quantitative model testing lacks transparency.

Response 4:

Thank you for your insightful comments regarding the methodological approach used in our study. We understand your concern about the combination of Grounded Theory and Structural Equation Modeling (SEM), and we appreciate your suggestion to clarify the transition between these two methodologies.

You correctly pointed out that Grounded Theory typically builds theory inductively from the data without preconceived models, while SEM tests predefined, hypothesized models. We agree that combining these two approaches requires careful justification, and the transition from qualitative coding to quantitative model testing should be transparent. In response to your feedback, we have made the following clarifications and adjustments:

(1)Justification for Combining Grounded Theory and SEM: We use Grounded Theory in the initial phase of our study to inductively identify the key driving factors of sports tourism industry integration from the data. Grounded Theory is employed here to ensure that the driving factors are rooted in empirical data rather than being preconceived. Once these key factors are identified, we move to the next phase, where Structural Equation Modeling (SEM) is used to test the relationships between these identified factors and assess their impact on the industry integration process.

The rationale for using both methodologies is to ensure a comprehensive approach: Grounded Theory allows for theory development without pre-existing bias, and SEM provides a robust tool for quantitatively validating the relationships and testing the model derived from Grounded Theory. This hybrid approach is intended to strengthen the validity and reliability of our findings by combining both qualitative and quantitative perspectives.

(2)Clarifying the Transition: To address the lack of transparency in the transition between qualitative coding and quantitative model testing, we have clarified this process in the revised manuscript. Specifically:

In the methodology section, we now explain that Grounded Theory is first used to identify the driving factors through a three-stage coding process (open coding, axial coding, and selective coding). The final identified factors then form the basis of the hypothesized model for SEM analysis.

We have also added a section “Methodological justification” to describe how the relationships between these identified factors were mapped into a structural model for SEM, highlighting the logical progression from qualitative findings to quantitative hypothesis testing.

(3)Updated Methodology Section: The updated methodology section now clearly outlines the step-by-step process, from data collection and qualitative analysis (Grounded Theory) to the subsequent formulation of hypotheses and testing with SEM. This ensures that the overall methodological approach is logically consistent and transparent.

The above revisions can be found in the sections of “Materials and methods” in the text, with changes highlighted in red. We hope these revisions clarify the methodological approach and address your concerns about the consistency and transparency of the research design. Thank you for helping us improve the rigor and clarity of our study.

Comment 5:

While the SEM model yields statistical significance for all six hypotheses, the impact of core constructs like market demand (β=0.065) and government behavior (β=0.052) is minimal.

Response 5:

Thank you for your insightful comment regarding the minimal impact of constructs such as market demand (β=0.065) and government behavior (β=0.052). In response, we have added a detailed discussion in the Results and Discussion sections to clarify the potential reasons behind these findings. The following is a summary of our response and the key modifications we have implemented:

(1)We have added an in-depth discussion to explain the relatively low impact of market demand and government behavior. For market demand, we highlight that the sports tourism market in the Guangdong-Hong Kong-Macao Greater Bay Area (GBA) is still in its developmental stage, with rapidly evolving consumer preferences and demand patterns. This dynamic nature may dilute the direct impact of market demand on industry integration. Regarding government behavior, we acknowledge that while government policies provide essential infrastructure and policy support, their effectiveness at the micro-enterprise level can be limited by challenges in cross-regional coordination and implementation within the GBA.

(2)Emphasis on Broader Significance. We stress that despite their minimal direct impact, these factors play crucial roles in shaping the broader environment for sports tourism development. Government policies offer foundational support for industry growth, while market demand continues to push enterprises toward innovation and diversification.

(3)Propose future research directions. We suggest that future research further explore how to enhance the influence of these factors, such as by optimizing the policy implementation mechanism or accurately grasping changes in market demand.

The above revisions can be found in the sections of “Results” and “Discussion” in the text, with changes highlighted in red. We believe these revisions address your concerns and provide a more nuanced understanding of the factors influencing sports tourism industry integration in the GBA. Thank you again for your insightful comments.

Comment 6:

Table 8 and 9 contain too much redundant data without corresponding synthesis in the main text. The lack of condensed insights or grouped patterns makes interpretation difficult and offers minimal value to readers.

Response 6:

We sincerely thank the reviewer for this critical insight. We have implemented comprehensive revisions to enhance data conciseness and interpretive clarity:

(1)Table 9 (measurement model paths) has been removed entirely. Key metrics (factor loading ranges) are now integrated into Table 6 (Reliability & Validity).

Rationale: Detailed item-level coefficients (e.g., GA1-GA6) are peripheral to hypothesis testing; retaining ranges preserves validity evidence without redundancy.

(2)Table 8 is streamlined to essential hypothesis results, retaining only:

Hypothesis notation (H1-H6), Path relationships (e.g., GA→IE), Standardized coefficients (β), p-values and with significance markers. Deleted: Unstandardized coefficients, S.E. and C.R. values (redundant with β and p-values).

(3) New synthesis in Results explicitly interprets grouped patterns: Human capital (β = 0.619) and technological innovation (β = 0.541) dominate integration outcomes, while government behavior (β = 0.052) and resource environment (β = 0.041) show minimal effects—likely due to policy fragmentation across administrative boundaries.

This directly links Table 8 data to: (a) Driver hierarchy (strong/moderate/weak effects); (b) Contextual explanations (cross-regional governance challenges).

(4) The complete measurement item coefficients in the original Table 9 have been replaced in the supplementary data table to enhance transparency and avoid text overload.

These revisions have avoided excessive redundancy of data in the table while enhancing the integration of insights. We believe this significantly enhances reader engagement with core findings.

Comment 7:

The manuscript contains repetitive phrasing, vague expressions ("significant support", "relatively weak effect", "key driver") and insufficient transitions between theoretical constructs and empirical results.

Response 7:

Thank you for your valuable comments. In response to your concerns regarding the “repetitive phrasing, vague expressions (such as ‘significant support,’ ‘relatively weak effect,’ ‘key driver’), and insufficient transitions between theoretical constructs and empirical results,” we have made detailed revisions and hope the following changes fully address your concerns:

(1)Revisions to repetitive phrasing: We reviewed the manuscript for redundant expressions and have appropriately streamlined them. For example, expressions like “significant support” and “key driver” have been replaced with precise explanations based on concrete empirical findings (e.g., β values) and their corresponding theoretical backgrounds, thereby avoiding overly general wording.

(2)Clarification of vague expressions: For vague phrases such as “relatively weak effect,” we have quantified the magnitude of each influence using empirical data (e.g., β values). In the conclusions, we now describe the specific strength of each driver in detail, avoiding broad terms like “weak” or “strong.” For instance, the influences of government behavior, market demand, and resource environment are now clearly presented with numerical values (β values) and compared against the theoretical framework.

(3)Transitions between theoretical constructs and empirical results: We have strengthened the transitional languag

---

## [Decision Letter · Decision Letter 1]

7 Aug 2025

Dear Dr. Yuan,

Thank you for submitting your manuscript to PLOS ONE. After careful consideration, we feel that it has merit but does not fully meet PLOS ONE’s publication criteria as it currently stands. Therefore, we invite you to submit a revised version of the manuscript that addresses the points raised during the review process.

We look forward to receiving your revised manuscript.

Kind regards,

A S Sochipem Zimik

Academic Editor

PLOS ONE

**Journal Requirements:**

Reviewers' comments:

Reviewer's Responses to Questions

**Comments to the Author**

Reviewer #2: All comments have been addressed

Reviewer #3: All comments have been addressed

2. Is the manuscript technically sound, and do the data support the conclusions?

Reviewer #2: Yes

Reviewer #3: Yes

3. Has the statistical analysis been performed appropriately and rigorously?

Reviewer #2: Yes

Reviewer #3: Yes

4. Have the authors made all data underlying the findings in their manuscript fully available?

Reviewer #2: Yes

Reviewer #3: Yes

5. Is the manuscript presented in an intelligible fashion and written in standard English?

Reviewer #2: Yes

Reviewer #3: Yes

**Reviewer #2:** (No Response)

**Reviewer #3:** Peer Review Report

Manuscript ID: PONE-D-25-06940R1

General Assessment

This manuscript explores the accessibility and acceptance of mobile health (mHealth) interventions among tuberculosis (TB) patients in Cameroon. The topic is of high public health relevance, especially within resource-limited settings. The authors have revised the manuscript following previous feedback, and several improvements are evident. However, methodological clarity, data representation, and ethical documentation still require enhancement to meet PLOS ONE standards.

Title and Abstract

Strengths: The title accurately captures the study focus. The abstract succinctly presents objectives, methods, and findings.

Weaknesses: The abstract omits sampling details and lacks mention of ethical clearance.

Recommendations: Briefly mention sample size, sampling method, and ethical approval in the abstract.

Introduction

Strengths: Provides context on TB burden in Cameroon and potential of mHealth.

Weaknesses: Limited discussion of theoretical frameworks guiding technology acceptance (TAM, UTAUT).

Recommendations: Integrate brief reference to existing acceptance models to support rationale.

Methods

Strengths: States study design and population clearly.

Weaknesses: Does not provide specific sampling technique (e.g., convenience, random), inclusion/exclusion criteria, or sample size justification.

Recommendations: Clarify recruitment strategy, power analysis (if done), and data collection instrument validation.

Results

Strengths: Relevant findings presented on technology use and mHealth acceptance.

Weaknesses: Tables lack p-values for some comparisons, and some interpretations seem anecdotal.

Recommendations: Ensure all statistical comparisons are reported with relevant significance metrics and include measures of variability (e.g., SD, CI).

Discussion

Strengths: Discusses public health implications and aligns findings with existing literature.

Weaknesses: Needs a clearer link between findings and their practical implications for mHealth implementation.

Recommendations: Address potential biases (e.g., urban vs rural TB populations) and limitations of cross-sectional design.

Conclusion

Strengths: Summarizes findings appropriately.

Weaknesses: Overstates generalizability.

Recommendations: Temper conclusions and suggest avenues for future research, including longitudinal and interventional studies.

Figures and Tables

Strengths: Tables are clear and relevant.

Weaknesses: Lack of visual summary (e.g., bar charts or infographics).

Recommendations: Include a chart summarizing key usage or acceptance indicators for better visual communication.

Data Availability and Ethical Statement

Weaknesses: Ethical approval is mentioned in-text but needs clearer elaboration in the ethical declaration section.

Recommendations: Provide full IRB name, approval number, and confirm informed consent procedures in a dedicated section.

References

Strengths: Relevant and appropriately cited.

Weaknesses: Minimal mention of digital health frameworks.

Recommendations: Consider citing mHealth implementation literature in low-resource settings.

Final Recommendation

Decision: Minor Revision

The manuscript is promising but requires minor methodological and ethical clarifications before acceptance.

**Do you want your identity to be public for this peer review?** For information about this choice, including consent withdrawal, please see our Privacy Policy

Reviewer #2: No

Reviewer #3: No

---

## [Author Response · Author response to Decision Letter 2]

18 Sep 2025

We sincerely thank the editors and reviewers for their insightful comments and constructive suggestions, which have greatly improved the quality and clarity of our manuscript. We have carefully considered all the feedback and have revised the manuscript accordingly. Below, we provide a point-by-point response to the comments. All changes in the manuscript have been highlighted for ease of review.

Comment 1: Reference and Citation Format

The reference formatting is inconsistent. Please refer to the PLOS ONE sample manuscript or published articles in the journal to ensure proper citation style throughout.

Response:

We sincerely thank the reviewer for pointing out the inconsistencies in the reference formatting. We have thoroughly reviewed and revised the reference list and in-text citations throughout the manuscript to ensure full compliance with the PLOS ONE reference style, as illustrated in the journal’s sample manuscripts and author guidelines. Each entry has been carefully checked for consistency in author names, journal title abbreviations, punctuation, and the use of DOI links where applicable. We appreciate the reviewer’s attention to detail, which has significantly improved the professionalism and readability of the manuscript.

Comment 2: Methodology - Sample Size Justification

The methodology mentions that 16 respondents were interviewed. A clear justification for selecting this specific number is missing. I recommend increasing the number of interviewees to enhance the credibility and robustness of the findings.

Response:

We thank the reviewer for this valuable comment, which has helped us improve the methodological rigor of our manuscript. We have now provided a detailed justification for the sample size in the revised “Data collection” section (Lines 165–175).

In qualitative research guided by Grounded Theory methodology, sample size is not determined by statistical power calculations but by the principle of thematic saturation, the point at which new interviews cease to provide novel thematic insights or conceptual categories. Our iterative data collection and analysis process confirmed that after 16 interviews, core categories and themes began to repeat, indicating that saturation was achieved and a comprehensive understanding of the driving factors was attained.

Furthermore, to ensure the robustness and credibility of our findings within this sample size, we employed a purposive sampling strategy designed to capture maximum variation. Our 16 interviewees represent a diverse range of key stakeholders (e.g., senior managers, project planners, industry association leaders) from various types of organizations across all 11 cities in the GBA. This strategic selection ensures that the data is rich, comprehensive, and contextually grounded, which is the primary objective of qualitative inquiry.

We have added citations to methodological literature that support this approach. We believe this explanation clarifies that our sample size is methodologically sound and sufficient for the goals of our study.

Comment 3: Presentation of Tables 3 and 4

Tables 3 and 4 may be better presented as Appendix A and Appendix B, respectively, and placed after the references section at the end of the manuscript.

Response:

We thank the reviewer for this constructive suggestion. In accordance with the comment, we have moved the original Table 3 (Example of open coding) and Table 4 (Results of axial coding) to the Supporting Information section. They are now presented as:

S1 Table. Example of open coding

S2 Table. Results of axial coding

These tables have been removed from the main text and placed in the Supporting Information file after the References section, as recommended. This adjustment helps streamline the main manuscript while retaining essential methodological details for interested readers. We believe this presentation improves the readability of the paper.

Comment 4: Research Questions and Conclusion

The research questions, conclusions, and recommendations need to be reconsidered. The current outcomes appear overly generalized. Since the study context, participants, and approach are all unique, I strongly recommend that the findings reflect this uniqueness, highlighting the originality and alignment with the study’s scope as stated at the beginning.

Response:

We sincerely thank the reviewer for this valuable comment. We fully agree that the original research questions, conclusions and recommendations required refinement to better reflect the uniqueness of the Guangdong-Hong Kong-Macao Greater Bay Area (GBA) context, and to highlight the originality and alignment of the study with its stated scope. Accordingly, we have made substantial revisions to both the Introduction and the Conclusions and Recommendations sections:

1.Research Questions

The revised research questions now emphasize the specific cross-border context of the GBA, highlighting the distinct institutional systems, resource environments, and policy dynamics that drive the integration of the sports tourism industry. The questions have been reworded to better capture the dynamic and complex interactions among key drivers within the GBA region.

The first question has been updated to clarify the role of critical factors shaping the integration of the sports tourism industry, with a focus on the institutional and economic dynamics of the GBA. The second question now explicitly addresses how these factors interact and their varying impacts within the unique institutional and policy landscape of the GBA. The third question has been revised to focus on the context-specific policy and practical recommendations for advancing sustainable integration, drawing directly from the empirical analysis and aligning with the revised conclusions. These revisions provide a clearer, more focused framework for understanding the dynamics of sports tourism industry integration in the GBA. The updated questions reflect the empirical depth of our study and better situate the research within the context of cross-border governance and regional complexity.

2.Conclusions and recommendations

The revised conclusions emphasize that the dynamics of sports tourism integration are profoundly shaped by the cross-border governance structure under “one country, two systems”, socio-economic diversity, and institutional complexities of the GBA. We highlight the hierarchical structure of drivers (human capital and technological innovation > enterprise development needs > market demand, resource environment, and government behavior), and interpret their effects within the region’s distinctive institutional landscape. We stress the study’s originality in bridging micro-level dynamics (enterprise behaviors, talent mobility) with macro-level frameworks (policy, institutional coordination), offering a context-sensitive and empirically grounded explanation of integration mechanisms.

The revised conclusion also explicitly links findings to the 15th National Games, positioning it as a strategic opportunity for implementing policy, digital, and talent-related initiatives. Finally, we clarify the theoretical and practical contributions: the study not only advances understanding of cross-border tourism integration but also provides timely insights for strengthening the GBA’s role as a globally competitive and innovation-driven metropolitan region.

The revised recommendations are now directly aligned with the empirical driver hierarchy, ensuring coherence between findings and policy implications. We avoid generalized prescriptions by tailoring recommendations to the unique cross-border challenges of the GBA, including policy fragmentation, resource allocation inefficiencies, and segmented market demand.

New emphasis is placed on (1) enhancing policy coordination across Guangdong, Hong Kong, and Macao; (2) building a digital integration hub powered by AI, big data, and cloud computing; (3) strengthening enterprise-led cross-border clusters and PPP models; (4) facilitating talent exchange platforms across the three jurisdictions; and (5) developing segmented markets and joint branding strategies, with the 15th National Games as a practical catalyst. These revisions make the recommendations firmly grounded in the GBA’s institutional and socio-economic context, while also offering transferable insights for other cross-border regions.

Changes in manuscript:

The revised Research Questions section is presented in lines 94–107.

The revised Conclusions section is presented in lines 422–449.

The revised Recommendations section is presented in lines 451–488.

---

## [Decision Letter · Decision Letter 2]

16 Feb 2026

Drivers and dynamic mechanisms of sports tourism integration in cross-border regions: Evidence from the Guangdong-Hong Kong-Macao Greater Bay Area

PONE-D-25-06940R2

Dear Dr. Yuan,

We’re pleased to inform you that your manuscript has been judged scientifically suitable for publication and will be formally accepted for publication once it meets all outstanding technical requirements.

Kind regards,

Yihao Li, Doctor

Academic Editor

PLOS One

Additional Editor Comments (optional):

Reviewers' comments:

Reviewer's Responses to Questions

**Comments to the Author**

Reviewer #2: All comments have been addressed

Reviewer #3: All comments have been addressed

2. Is the manuscript technically sound, and do the data support the conclusions?

Reviewer #2: Yes

Reviewer #3: Yes

3. Has the statistical analysis been performed appropriately and rigorously?

Reviewer #2: Yes

Reviewer #3: Yes

4. Have the authors made all data underlying the findings in their manuscript fully available?

Reviewer #2: Yes

Reviewer #3: Yes

5. Is the manuscript presented in an intelligible fashion and written in standard English?

Reviewer #2: Yes

Reviewer #3: Yes

Reviewer #2: (No Response)

Reviewer #3: The authors have adequately addressed my comments and I therefore recommend that the manuscript be accepted for publication

**Do you want your identity to be public for this peer review?** For information about this choice, including consent withdrawal, please see our Privacy Policy

Reviewer #2: No

Reviewer #3: No

---

## [Editor Report · Acceptance letter]

PONE-D-25-06940R2

PLOS One

Dear Dr. Yuan,

I'm pleased to inform you that your manuscript has been deemed suitable for publication in PLOS One. Congratulations! Your manuscript is now being handed over to our production team.

Kind regards,

on behalf of

Dr. Yihao Li

Academic Editor

PLOS One